# Is anyone truly healthy? Trends in health risk factors prevalence and changes in their associations with all-cause mortality

**Winnie W. Yu, Rubin Pooni, Chris I. Ardern, Jennifer L. Kuk**⊙\*

School of Kinesiology and Health Science, York University, Toronto, Canada

\* jennkuk@yorku.ca

## Abstract

### Objective

The purpose of the study was to determine trends in the prevalence of individual health risk factors across time and to examine if their associations with mortality have changed over time.

### Methods

Data from the National Health and Nutrition Examination Surveys (NHANES III– 1988–1994 and NHANES 1999–2014; age ≥20 years) was used to examine differences in the odds ratio (OR) of 5-year mortality risk associated with various common health risk factors over the two survey periods using weighted logistic regression analysis adjusting for age, sex, obesity category and white ethnicity (n = 28,279).

### Results

Over 97% of individuals had at least one of the 19 risk factors examined with no difference in the prevalence over time (P>0.34). The prevalence of lifestyle, social/mental and physical risk factors (2.2 to 19.1%) increased over time (P<0.0002), while the prevalence of having physiological risk factors decreased by ~6.5% (P<0.0001). Having any lifestyle or social/mental risk factor was significantly associated with a higher 5-year OR for mortality risk in 1999–2014, than 1988–94. In particular, having low education or use of mental health medication were not associated with mortality risk in 1988–94 (P>0.1), but were significantly associated with a higher 5-year OR for mortality in 1999–2014 (P<0.0001). Conversely, physiological risk factors were more weakly related with mortality risk in 1988–1994, than 1999–2014. Having any physical risk factor, and poor self-rated health were similarly related with 5-year mortality risk at both timepoints.

### Conclusion

Health risk factors have both increased and decreased in prevalence over time, along with changes in the association between many of the risk factors and mortality risk. Taken

**Funding:** The author(s) received no specific funding for this work.

**Competing interests:** The authors have declared that no competing interests exist.

together, these changes complicate interpretation of temporal trends and warrant cautious interpretation of population health patterns based on surveillance data.

## Introduction

Surveillance of known risk factors has been a common aim for public health and research. Cardiovascular disease (CVD) and cancer have consistently been the leading causes of mortality in the U.S. [1], though the prevalence of CVD mortality has reduced over time [2]. It is well known that lifestyle and social factors are associated with CVD and cancer risk [3]. However, little attention to date has been paid to the potential changing association between these risk factors and morbidity and mortality over time. Indeed, changes in lifestyle behaviours in addition to the built environment, health care or social programs can all influence many dimensions of chronic health risk [4]. For example, there have been decreases in the prevalence of CVD risk over time which may be reflective of the improvements in medications and treatments for CVD [5]. These improvements in health care may have made lifestyle behaviours, such as physical activity and diet, less beneficial over time. Conversely, over a similar time period, the prevalence of new cancer cases has increased [3], while cancer mortality rates have declined [6]. Over time, there have also been improvements in the stigma of mental health and addiction, and public health promotion efforts to decrease smoking [7] and alcohol abuse in youth [8]. Decreases in stigma and/or later adoption of drug use may result in better care and lower mortality risk.

Thus, the objective of this study was to examine the changes in the prevalence of various health risk factors and their association with mortality risk over time.

## Methods

### Study population

The current study is a cross-sectional comparison of two nationally representative samples from 1988–1994 versus 1999–2014 with 5-year mortality follow-up. Participant data was obtained from the publicly available National Health and Nutrition Examination Surveys (NHANES) III (n = 33,994) and continuous between 1999 and 2014 (n = 82,901), so that 5-year mortality follow-up would be available. All survey participants gave written informed consent. NHANES III is a nationally representative cross-sectional survey that was conducted between 1988 and 1994 by the National Center for Health Statistics of the Centers for Disease Control and Prevention, in persons aged 2 months or older. The Continuous NHANES are a series of nationally representative cross-sectional surveys that are released biannually starting in 1999. As this is an analysis of publicly available data, the current study did not require ethics approval from our institutional review board.

Both surveys are collected using a stratified, multistage, probability cluster design of the non-institutionalized U.S. population. Data was collected at home interviews and health examinations at Mobile Examination Centers (MEC). Details regarding study design, protocols, laboratory and clinical measurements and analytical guidelines have been previously published elsewhere [9–11].

Participants were included in the analytical sample if they were 20 years or older (n = 62,575). Participants were further excluded if they were pregnant or had missing body mass index (BMI) leaving 48,003 participants. Participants with missing data for lifestyle factors (physical activity; high fat intake; smoking status; alcohol intake), physiological health risk

factors (high blood pressure; hyperglycemia; dyslipidemia; CVD; cancer; lung problems), social/mental factors (lack of health insurance; low education; food insecurity; low income; use of mental health medications), physical factors (pain medications; arthritis; limitations of activities of daily living (>60 yr of age only); obesity) and general health status were excluded from the analytical dataset, leaving 28,306 participants.

To make the follow-up more comparable between NHANES III and continuous, five-year mortality status was determined using public access National Center for Health Statistics (NCHS) Mortality Linkage Files with follow-up to December 31 at years 2006, 2011, 2015 and 2019. Follow-ups were truncated at 5 years. Deaths that occurred after a follow-up of greater than 5 years were recoded as censored events. Individuals with less than 5-year follow-up were excluded from the analytical dataset, leaving 28,279 participants.

## Defining health risk factors

Health risk factors were chosen because they are common health risk factors that were consistently available across all survey years. Risk factors are classified as the absence (0) or presence (1) of any of the risk factors within these 4 categories:

Lifestyle–no physical activity; high fat diet; current smoking; alcohol consumption

Physiological–high blood pressure; hyperglycemia; dyslipidemia; CVD; cancer; lung problems

Social/Mental–lack of health insurance; low education; food insecurity; low income; use of mental health medication

Physical–arthritis; use of pain medication; obesity; limitations of activities of daily living (> 60 only).

General self-rated health reported as: excellent, very good, good, fair and poor, was re-categorized as poor or not poor self-rated health.

## Lifestyle risk factors

Physical activity was self-reported as the amount of moderate/vigorous leisure time physical activity over the past month. Those reporting none were categorized as 'No Physical Activity'. High fat diet was classified as a fat consumption of greater than 35% of total calories from the diet records. Current smoking status was assessed by self-report. Excessive alcohol consumption was classified as consuming greater than an average of 2 drinks on the days they consume alcohol for men and greater than 1 drink per day for women over the past 12 months [12].

## Physiological risk factors

Blood pressure, fasting glucose, triglycerides, cholesterol, LDL and HDL were assessed at the mobile exam center and analyzed using standard methods [13]. High blood pressure was classified as measured blood pressure > = 140/90 mmHg [14], self-report diagnosed hypertension or use of hypertensive medication. High lipid was classified as measured fasting cholesterol > = 5.2 mM [15], triglycerides > = 2.0 mM [15], self-report diagnosed high cholesterol or use of lipid medication. High fasting glucose was classified as measured blood glucose > = 6.1 mM [16], self-report diagnosed diabetes or use of diabetic medication. Self-reported CVD was classified as ever having myocardial infarction, coronary or congenital heart disease, or stroke, or reported use of CVD medications. History of any cancer was assessed by self-report. Lung disease was self-reported history of emphysema or chronic bronchitis.

### Social/Mental risk factors

Participants were asked about their access to health insurance (none versus any), highest educational attainment (less than high school versus high school or more), income (Poverty Income Ratio (PIR) > = 1.3) [17] and food insecurity (any food insecurity versus none). Use of mental health medications was considered as self-reported use of anxiety, antipsychotic or depression medications.

### Physical risk factors

Limitations in activities of daily living (ADL) were assessed in adults over 60 years of age [18] and were classified as difficulties with any of the following: walking for a quarter mile; walking up ten steps; stooping, crouching or kneeling; lifting or carrying; house chores; preparing meals; managing money; walking between rooms on the same floor; standing up from an armless chair; getting in and out of bed; using a fork, knife or drinking from a cup; or dressing themselves. This list was used as these ADLs were asked in all surveys.

Individuals were asked to self-report arthritis and use of pain medication. Obesity was classified as having a body mass index (BMI) over 30 kg/m$^2$ [19]. Measured height and weight were assessed using standard protocols [20].

### Statistical analysis

Prevalence for each health risk factor and category were determined for NHANES III and averaged over four-year periods for NHANES continuous to improve the stability of the estimated prevalence, particularly for risk factors that were more rare. Linear regression was used to determine differences in prevalence over time within the samples combined. Logistic regression models were used to estimate the 5-year odds ratios (OR) with 95% confidence intervals for all-cause mortality with adjustment for age, sex (male versus female), BMI category (underweight, normal weight, overweight and obesity) and white ethnicity separately within the NHANES III and Continuous cycles. Estimates stratified by survey (i.e., NHANES III and continuous separately) were weighted to be nationally representative. For analyses comparing the two surveys (NHANES III and continuous differences), it was not possible to weight the analysis to be nationally representative. Statistical analyses were performed using the SAS statistical software (version 9.4; SAS Institute, Cary, NC) with significance defined at p<0.05.

### Results

Characteristics of the NHANES III and continuous surveys 1999–2014 are presented in **Table 1**. The weighted prevalence of all 19 risk factors were categorized into 4 categories: Lifestyle, Physical, Physiological, and Mental/Social. Changes in prevalence over time was noted in 18 of the 19 health risk factors in men and/or women (P<0.05, **Table 1**). Over 97% of individuals had at least one of the 19 risk factors at all time points with no difference in the prevalence of having any risk factor over time (P>0.34, M: 98.1 to 98.8%; F: 98.1 to 97.5%). Self-reported 'poor' health subtly decreased in prevalence over the study period in both men and women (P<0.02, M: 14.3 to 13.9%; F: 16.3 to 14.2%), with no sex difference.

**Figs 1 and 2** show the differences in the prevalence of the various risk factor categories and individual risk factors. The prevalence of lifestyle risk factors (M: 81.3 to 83.6%; F: 77.8 to 86.2%), social/mental risk factors (M: 38.1 to 48.7%; F: 38.3 to 51.6%) and physical factors (M: 33.3 to 52.5%; F: 43.9 to 59.0%) increased between 1988–1994 to 2011–2014 (P<0.0002), while the prevalence of having physiological risk factors decreased (M: 85.2 to 78.6%; F: 85.7 to 79.3%, P<0.0001) over the same time frame. Overall, having lifestyle and physical factors were

**Table 1. Participant characteristics between 1988–94 to 2011–14 in men and women.**

| Men | 1988–94 | 99–02 | 03–06 | 07–10 | 11–14 | p Trend over Time |
|---|---|---|---|---|---|---|
| Age | 43.8 (0.5) | 42.5 (0.4) | 43.7 (0.6) | 44.6 (0.4) | 49.4 (0.6) | 0.07 |
| BMI | 26.5 (0.1) | 27.4 (0.1) | 28.2 (0.2) | 28.6 (0.2) | 28.7 (0.2) | <0.0001 |
| Health Factor (#) | 2.4 (0.1) | 2.4 (0.1) | 2.5 (0.1) | 2.6 (0.1) | 2.6 (0.1) | <0.0001 |
| **Health Risk factor (%)** | | | | | | |
| Lifestyle | 81.3 (1.1) | 81.7 (1.3) | 81.1 (0.8) | 84.0 (1.0) | 83.6 (1.0) | <0.0001 |
| Physiological | 85.2 (1.3) | 75.4 (1.3) | 74.0 (1.2) | 75.8 (0.9) | 78.6 (1.4) | <0.0001 |
| Social/Mental | 38.1 (1.8) | 45.1 (1.9) | 45.3 (1.6) | 46.7 (1.5) | 48.7 (2.1) | 0.0002 |
| Physical | 33.3 (1.2) | 38.8 (1.1) | 47.4 (1.6) | 49.4 (1.5) | 52.5 (1.6) | <0.0001 |
| No Risk Factors | 1.9 (0.4) | 2.7 (0.6) | 2.4 (0.4) | 2.1 (0.4) | 1.2 (0.3) | 0.82 |
| **Women** | **1988–94** | **99–02** | **03–06** | **07–10** | **11–14** | **p Trend over Time** |
| Age | 44.8 (0.6) | 43.4 (0.4) | 44.8 (0.5) | 45.2 (0.6) | 50.3 (0.6) | 0.03 |
| BMI | 26.4 (0.2) | 27.6 (0.2) | 27.9 (0.2) | 28.1 (0.1) | 29.0 (0.2) | <0.0001 |
| Health Factor (#) | 2.5 (0.1) | 2.5 (0.1) | 2.6 (0.1) | 2.6 (0.1) | 2.8 (0.1) | <0.0001 |
| **Health Risk Factor (%)** | | | | | | |
| Lifestyle | 77.8 (1.1) | 84.4 (1.1) | 83.8 (1.2) | 87.5 (1.0) | 86.2 (1.3) | <0.0001 |
| Physiological | 85.7 (1.1) | 76.2 (1.1) | 73.3 (1.4) | 74.3 (0.9) | 79.3 (1.3) | <0.0001 |
| Social/Mental | 38.3 (1.7) | 45.0 (1.8) | 47.6 (1.9) | 50.8 (1.2) | 51.6 (2.4) | <0.0001 |
| Physical | 43.9 (1.5) | 48.6 (1.2) | 53.7 (1.1) | 51.7 (1.2) | 59.0 (1.6) | <0.0001 |
| No Risk Factors | 1.9 (0.4) | 2.5 (0.5) | 1.8 (0.3) | 1.4 (0.3) | 2.5 (0.4) | 0.34 |

Values presented are weighted means (SE). Differences by survey are tested using unweighted analyses (P<0.05).

more common in women than men (P<0.03), while there were no sex differences for social/ mental, physiological or any health risk factors (P>0.05).

Within lifestyle factors, lack of physical activity and excessive alcohol consumption increased in prevalence (P<0.0001), while smoking decreased in prevalence in men and women (P<0.02) and the prevalence of consuming of a high fat diet decreased from 49 to 44% in only men (P<0.0001, **Fig 1**).

Within physiological risk factors, having high blood pressure or glucose or cancer increased in prevalence, while having high lipids decreased in prevalence (**Fig 1**). The presence of CVD tended to decrease between NHANES III (1998–1994) and NHANES continuous 1999–2000 and then increased, but was generally lower in 2011–2014 than NHANES III in women, while lung disease did not change in prevalence over time (P>0.05).

For social/mental risk factors, there was an increase in the prevalence of individuals without health insurance, with food insecurity and taking mental health medications (**Fig 2**, P<0.0001), while there was a decrease in the prevalence of those with less than a high school education (P<0.0001). There was a modest but significant decline in the prevalence of low income in females (P<0.0001), but no significant change in men over time (P = 0.3).

For physical risk factors (**Fig 2**), use of pain medications, arthritis and obesity all significantly increased in prevalence over time in both men and women, while there was a decrease in ADL problems in adults over 60 years of age (P<0.05).

Over the 5-year follow-up there were 1319 deaths. The 5-year odds ratios for all-cause mortality are presented in **Table 2.** There were no deaths in those without any of the 19 risk factors. As compared to NHANES III (1988–1994), having any risk factors in NHANES continuous was associated with 30% lower odds of 5-year mortality (OR = 0.71, 0.6–0.8; P<0.0001).

Having any lifestyle risk factor was significantly associated with increased 5-year OR for mortality risk in 1999–2014, but not 1988–94 (**Table 2**). Specifically, the 5-year OR for

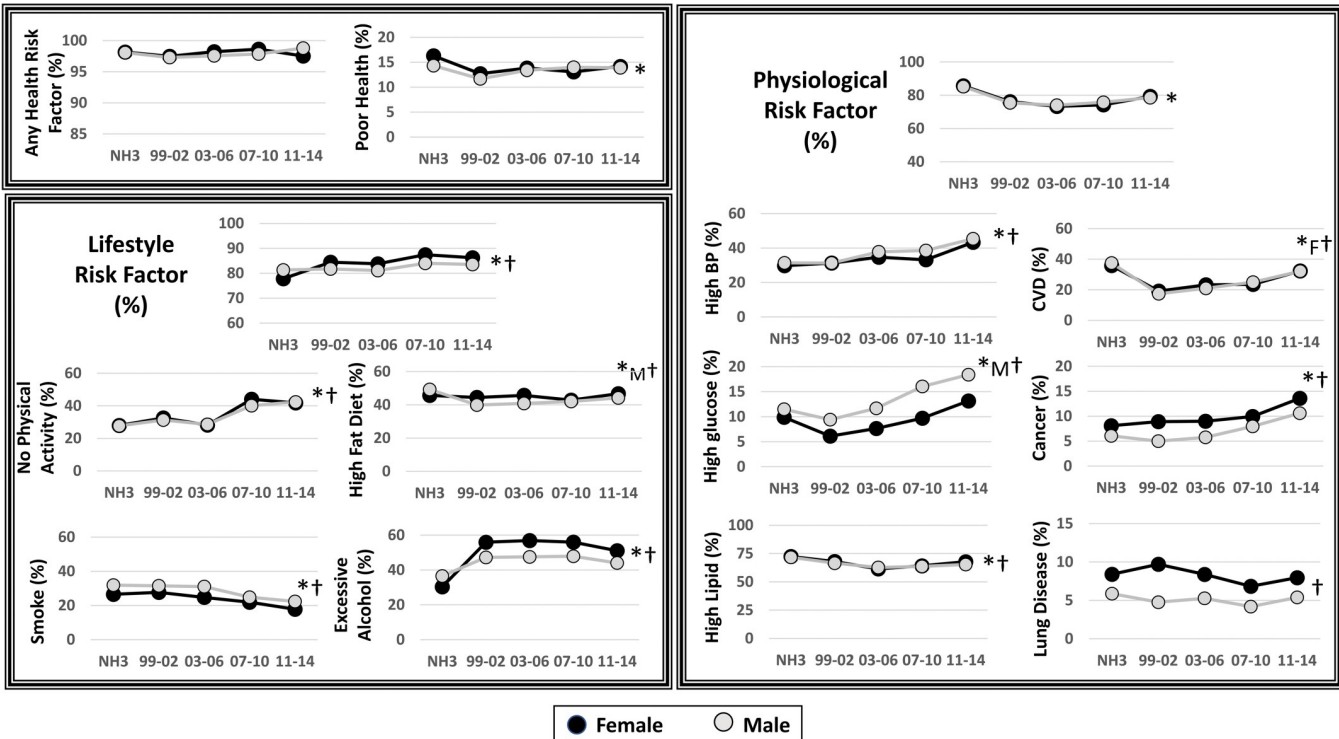

**Fig 1. Changes in the prevalence of having any health risk factors, poor self-rated health, lifestyle risk factors and physiological risk factors between 1988–1994 (NHANES III) and 2011–14 (NHANES continuous).** Each prevalence datapoint is weighted to be nationally representative. NH3 = NHANES III (1988–1994). BP = Blood Pressure; CVD = Cardiovascular Disease.* Significant trend over time were examined with unweighted analyses (P<0.05). † Sex main effect (P<0.05).

mortality risk were all higher for lack of exercise, smoking and excessive alcohol consumption in 1999–2014 than 1988–94. Consumption of a high fat diet was not associated with 5-year mortality risk at either time point.

Social and mental factors were also associated with increased mortality risk at both time-points (P≤0.0001, **Table 2**), with a subtly higher OR in the later survey years. In particular, having low education or use of mental health medication were not associated with mortality risk in 1988–94 (P>0.1), but were significantly associated with a higher 5-year OR for mortality in 1999–2014 (P<0.0001).

Having any physiological risk factor was strongly associated with increased mortality risk in 1988–1994, but was more weakly related in 1999–2014 (P<0.02; **Table 2**). Most of the physiological risk factors were similarly related with mortality risk at both timepoints, though they generally tended to have a lower 5-year OR for mortality at the later timepoint.

Having any physical risk factor, the individual physical risk factors and poor self-rated health were similarly related with 5-year mortality risk at both timepoints (**Table 2**).

## Discussion

The prevalence of individuals who are free of all of the 19 examined risk factors was less than 3%, at all time points. The prevalence of the various risk factors varied in the absolute prevalence and changes over time. The prevalence of poor self-rated health was generally ~15% across the study period. The 5-year mortality risk associated with these risk factors also varied over time. This demonstrates that there may not only be changes in the prevalence of risk factors, but also differences in how certain risk factors relate with mortality risk over time.

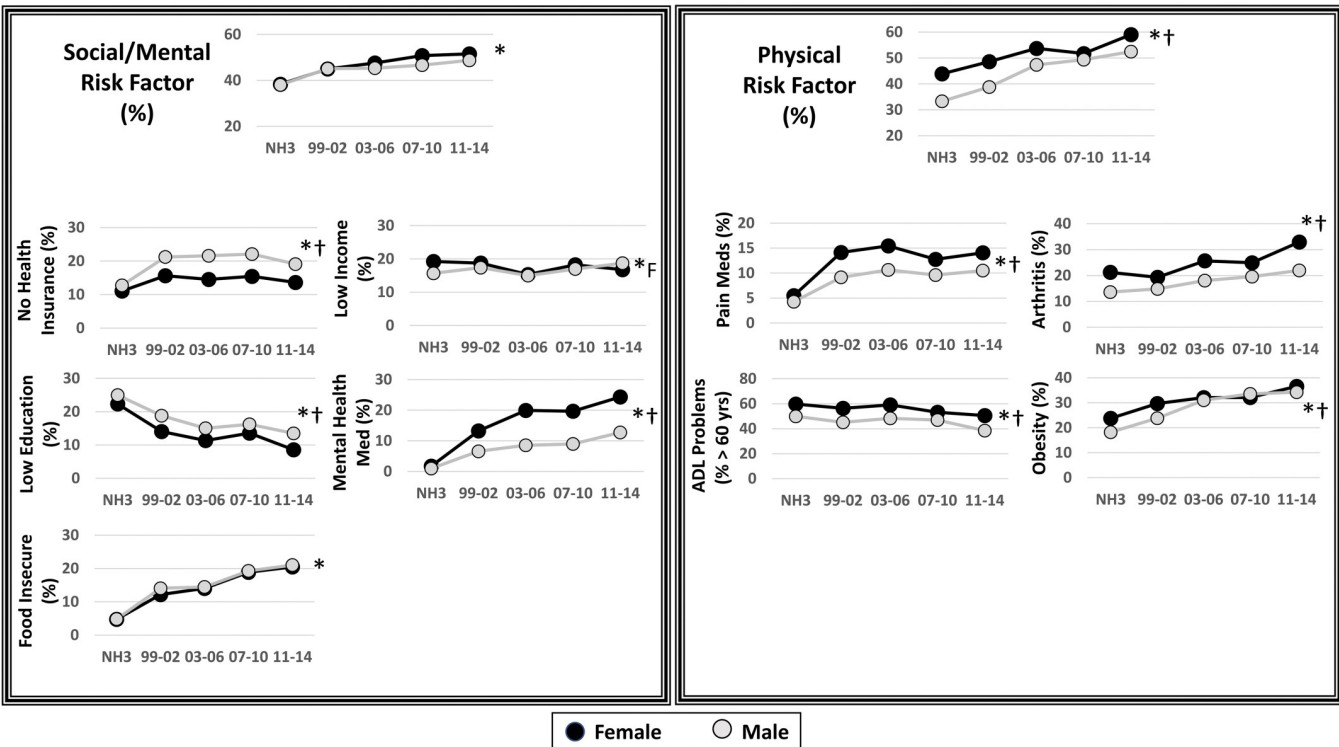

**Fig 2. Changes in the prevalence of having any Social/Mental and physical risk factors between 1988–1994 (NHANES III) and 2011–14 (NHANES continuous).** Each prevalence datapoint is weighted to be nationally representative. NH3 = NHANES III (1988–1994). ADL = Activities of Daily Living. * Significant trend over time were examined with unweighted analyses (P<0.05). † Sex main effect (P<0.05).

Public health has had a long focus on health promotion that has targeted lifestyle behaviours, such as physical activity, diet, smoking and excessive alcohol use [3], with variable success over time. Tobacco control has been one of the more successful programs [7] and the current study observed that the prevalence of smoking decreased from 32 to 22% in men and 27 to 18% in women. On the other hand, the prevalence of physical inactivity and excessive alcohol consumption has increased by 15% and 10%, respectively, since NHANES III (1988–94), while the prevalence of low fat diets were relatively consistent over the study period (~45%). The popularity of low fat diets first began in the 1960s when the link between high fat diets and CVD was first discovered [21]. Of late, the wisdom of prescribing low fat diets for health and particularly for obesity management has been called into question [22]. In this study, consuming a high fat diet was the only examined lifestyle factor that was not significantly associated with mortality risk at either time point. This may be due, in part, to differences in the association between high saturated versus high unsaturated fat with CVD [23]. Further, improvements in the treatment of CVD have likely reduced the negative impact of high fat diets [24, 25]. In fact, the association between cardiovascular risk factors (i.e., high blood pressure, high glucose, high lipid) and prevalent CVD with 5-year mortality risk was reduced over time. This is fortunate given the increased prevalence of high blood pressure and high glucose. Despite the higher rate of some CVD risk factors, there is a decreasing rate of premature heart disease mortality [25]. Nevertheless, heart disease is still the leading cause of death in the United States [26], and thus, understanding changes in the way risk factors relate with mortality risk over time warrants further investigation.

Many studies in the literature report that men consume more servings of alcohol than women and are more likely to have excessive consumption [27]. However, the dietary

**Table 2. Odds ratios for 5-year mortality risk in 1988–94 and 1999–2014.**

|  | NH3 1988–94 5 yr OR (95% CI) | P | NHC 1999–2014 5 yr OR (95% CI) | P |
|---|---|---|---|---|
| Lifestyle Factors | 1.41 (1.0–2.0) | 0.07 | 2.63 (2.0–3.5) | < .0001 |
| No Exercise | 1.92 (1.4–2.6) | 0.0001 | 2.53 (2.1–3.1) | < .0001 |
| High Fat Diet | 1.18 (0.9–1.5) | 0.22 | 0.93 (0.8–1.1) | 0.44 |
| Smoking | 1.69 (1.2–2.5) | 0.01 | 2.41 (1.9–3.1) | < .0001 |
| Excessive Alcohol | 0.97 (0.6–1.5) | 0.87 | 1.39 (1.1–1.8) | 0.01 |
| Social/Mental Factors | 1.66 (1.3–2.2) | 0.001 | 2.20 (1.8–2.7) | < .0001 |
| No Health Insurance | 2.02 (0.8–5.2) | 0.14 | 1.85 (1.3–2.7) | 0.001 |
| Low Education | 1.27 (0.9–1.7) | 0.13 | 1.99 (1.5–2.6) | < .0001 |
| Food Insecure | 2.44 (1.3–4.4) | 0.004 | 2.27 (1.8–2.9) | < .0001 |
| Low Income | 2.03 (1.4–3.0) | 0.0004 | 2.55 (2.1–3.1) | < .0001 |
| Mental Health Med | 0.87 (0.3–2.4) | 0.78 | 1.65 (1.3–2.1) | < .0001 |
| Physiological Factors | 3.52 (1.6–7.6) | 0.002 | 1.65 (1.1–2.5) | 0.02 |
| Cancer | 1.40 (1.0–2.0) | 0.07 | 1.49 (1.2–1.8) | 0.0003 |
| CVD | 2.12 (1.6–2.8) | < .0001 | 1.74 (1.4–2.2) | < .0001 |
| Lung Disease | 2.33 (1.6–3.5) | < .0001 | 2.07 (1.6–2.7) | < .0001 |
| High BP | 1.96 (1.4–2.7) | 0.0001 | 1.69 (1.3–2.1) | < .0001 |
| High Glucose | 2.03 (1.4–2.9) | 0.0003 | 1.64 (1.3–2.0) | < .0001 |
| High Lipid | 1.43 (1.0–2.1) | 0.07 | 0.97 (0.8–1.2) | 0.77 |
| Physical Factors | 1.93 (1.4–2.7) | 0.001 | 1.88 (1.4–2.5) | < .0001 |
| Arthritis | 1.14 (0.8–1.7) | 0.50 | 1.28 (1.0–1.6) | 0.02 |
| Pain Medication | 0.82 (0.5–1.4) | 0.46 | 1.19 (0.9–1.5) | 0.16 |
| ADL Problem | 2.00 (1.4–2.9) | 0.0003 | 2.13 (1.6–2.8) | < .0001 |
| ADL Problem (>60 yr) | 2.3 (2.5–3.5) | 0.0002 | 2.47 (1.9–3.3) | < .0001 |
| Obesity | 1.05 (0.8–1.4) | 0.77 | 1.15 (1.0–1.4) | 0.10 |
| Poor Health | 2.97 (2.1–4.2) | < .0001 | 3.28 (2.6–4.1) | < .0001 |

ADL = Activities of Daily Living; BP = Blood Pressure; CVD = Cardiovascular Disease

Models were adjusted for age, sex, BMI category and white ethnicity, and weighted within each survey separately.

guidelines for Americans [12] suggest that women should consume less servings of alcohol when they drink due to their smaller size and slower alcohol clearance [28]. Thus, in our study that used sex-specific cut-offs for alcohol, we observed that excessive alcohol was more prevalent in women, and has increased over time. Excessive alcohol consumption has been cited to be associated with increased mortality risk, CVD, cancer and injuries [29]. In the current study, excessive alcohol consumption was associated with higher mortality risk in 1999–2014, but not in 1988–1994. This is problematic when you consider the increased mortality risk in conjunction with the increased prevalence. Reasons for the changes in the pattern of alcohol consumption or the association with mortality risk is unclear, but may reflect differences in the types of alcohol consumed over time or perhaps differences in the reasons why individuals consume alcohol. Though not examined in this study, this may reflect greater incidence of negative health impacts of binge drinking or alcohol addiction [30, 31]. Alcohol is also a common coping mechanism for stress [32], and thus, excessive alcohol intake may be an indicator of other issues that contribute to poor health, particularly in later years. In fact, some more recent guidelines suggest even lower limits of no more than 2 servings per week [33], and that for cancer, there may be no safe limits for alcohol consumption [34, 35]. More work is needed to clarify the optimal amounts of alcohol consumption and health and to determine whether there are changes over time.

Use of mental health medications was also only significantly associated with 5-year mortality risk at the second survey, indicating a greater mortality risk associated with the use of mental health medications over time. This may be, in part, due to the overall low prevalence (<2%), and thus, low number of deaths in those taking mental health medications in 1988–1994 versus 1999–2014. However, individuals taking mental health medications are also more likely to have low education [36] and smoking [37], both factors which were also more strongly associated with mortality risk over time. These changes may reflect changes in access to care or perhaps decreases in stigma around mental health issues [38] and the importance of seeking treatment [39]. However, these changes in mortality risk may also suggest that the severity and negative effects of mental health issues may be more detrimental over time. Despite improvements in stigma, there is still significant stigma within health care that can lead to delays in the diagnosis and treatment of non-mental health conditions in patients with mental health disease [39]. A recent systematic review suggests that there are suboptimal adoption of the clinical guidelines [40]. Thus, it appears that improvements in care and treatment for mental health conditions may be needed.

The most dramatic difference in mortality risk was the decreased strength of association between having any physiological risk factor and mortality risk. Each of the cardiometabolic risk factors and lung disease each had modestly lower ORs for 5-year mortality risk in 1999–2014 than 1988–1994. These observations mirror other studies that report a decreasing prevalence of CVD mortality in the U.S. [41]. Improvements in CVD prevention and treatment are likely contributors to this improvement. There are concerns that declines in CVD may be derailed by the increasing prevalence of diabetes in the U.S. [41]. However, we observed that the association between diabetes and 5-year mortality risk is also decreasing in strength. Thus, the increased prevalence of risk factors in the population, may or may not translate into greater population mortality burden.

## Strengths and limitations

Strengths and limitations of the current study warrant mention. NHANES is designed to be representative of the U.S. population, and thus, is ideal for examining changes in the prevalence of health risk factors over time. One of the primary assumptions of mortality analyses using cox proportional hazards is that the relative risk is constant over time [42]. However, our study demonstrates that this assumption may not hold true for all health risk factors. One limitation is that the reference groups for our analyses differed between time points. Thus, the risk ratios are a reflection of changes in the mortality risk for those with and without the condition examined. Nevertheless, these analyses are still reflections of the associations of the risk factors within each timepoint examined, with some analyses changing in their statistical significance. Further studies should examine how the clustering of these risk factors act together in changing mortality risk over time, and perhaps sub-populations wherein these observations may differ.

## Conclusions

This study demonstrates that developments in health care, social programs and the built environment may have altered how traditional risk factors relate to health and mortality over time. Thus, attention to not only the changing prevalence of traditional risk factors, but also examination of how these risk factors change in their association with morbidity and mortality is needed.

## Author Contributions

**Conceptualization:** Winnie W. Yu, Rubin Pooni, Jennifer L. Kuk.

**Formal analysis:** Winnie W. Yu, Jennifer L. Kuk.

**Methodology:** Winnie W. Yu, Chris I. Ardern.

**Project administration:** Rubin Pooni, Jennifer L. Kuk.

**Supervision:** Chris I. Ardern, Jennifer L. Kuk.

**Writing – original draft:** Winnie W. Yu.

**Writing – review & editing:** Winnie W. Yu, Rubin Pooni, Chris I. Ardern, Jennifer L. Kuk.

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
