## [Decision Letter · Decision Letter 0]

2 May 2023

PONE-D-23-10979Is anyone truly healthy? Trends in health risk factors prevalence and changes in their associations with all-cause mortalityPLOS ONE

Dear Dr. Kuk,

Thank you for submitting your manuscript to PLOS ONE. After careful consideration, we feel that it has merit but does not fully meet PLOS ONE’s publication criteria as it currently stands. Therefore, we invite you to submit a revised version of the manuscript that addresses below points raised during the review process.

 Excellent study with good practical implications. a)In methods section of abstract: Can you elaborate further on what specific methods were used? b)In methods section of the main article: Can you specify what kind of study this, currently you just mentioned that SPSS software was used, that we used ODDS ratio. May be a line indication what kind of observational study this is?  c)At the end of the article, can we have separate brief conclusion section: include a take away point in the end d)Also may be a separate brief strength and limitation section? Please submit your revised manuscript by Jun 16 2023 11:59PM. If you will need more time than this to complete your revisions, please reply to this message or contact the journal office at plosone@plos.org. Please include the following items when submitting your revised manuscript:A rebuttal letter that responds to each point raised by the academic editor and reviewer(s). You should upload this letter as a separate file labeled 'Response to Reviewers'.A marked-up copy of your manuscript that highlights changes made to the original version. You should upload this as a separate file labeled 'Revised Manuscript with Track Changes'.An unmarked version of your revised paper without tracked changes. You should upload this as a separate file labeled 'Manuscript'.If applicable, we recommend that you deposit your laboratory protocols in protocols.io to enhance the reproducibility of your results. Protocols.io assigns your protocol its own identifier (DOI) so that it can be cited independently in the future. For instructions see: https://journals.plos.org/plosone/s/submission-guidelines#loc-laboratory-protocols. Additionally, PLOS ONE offers an option for publishing peer-reviewed Lab Protocol articles, which describe protocols hosted on protocols.io. Read more information on sharing protocols at https://plos.org/protocols?utm_medium=editorial-email&utm_source=authorletters&utm_campaign=protocols.

We look forward to receiving your revised manuscript.

Kind regards,

Vikramaditya Samala Venkata

Academic Editor

PLOS ONE

Journal Requirements:

Additional Editor Comments:

Good study, recommend edits as noted below

In abstract: In methods section, please describe more about what methods were used. Its too vague

In main article: In methods section, would it be possible to describe clearly what type of study this is. What kind of observational study this is? Cohort study?

End of discussion, can we have a separate conclusion section and a separate strengths/limitation section?

After above revision. Manuscript will be ready for publication

Reviewers' comments:

Reviewer's Responses to Questions

**Comments to the Author**

1. Is the manuscript technically sound, and do the data support the conclusions?

Reviewer #1: Yes

Reviewer #2: Yes

Reviewer #3: Yes

2. Has the statistical analysis been performed appropriately and rigorously? 

Reviewer #1: I Don't Know

Reviewer #2: Yes

Reviewer #3: Yes

3. Have the authors made all data underlying the findings in their manuscript fully available?

Reviewer #1: Yes

Reviewer #2: Yes

Reviewer #3: Yes

4. Is the manuscript presented in an intelligible fashion and written in standard English?

Reviewer #1: Yes

Reviewer #2: Yes

Reviewer #3: Yes

5. Review Comments to the Author

Reviewer #1: The authors have provided an analysis and reviewed data on the prevalence of various health risk factors over a period of nearly three decades in order to identify patterns in population health. By analyzing trends in the prevalence of these risk factors, they likely sought to determine how the health of the population has changed over time.

The examples of health risk factors that the authors have analyzed include smoking, alcohol consumption, physical inactivity, poor nutrition, and obesity amongst other social, physiological risk factors.

By examining changes in the prevalence of these risk factors over time, the authors have been able to identify trends and patterns that could inform public health policy and interventions aimed at improving population health.

Overall, it appears that the authors have conducted a comprehensive analysis of health risk factors over a long period of time in order to gain insights into population health patterns. The findings of this analysis could have important implications for efforts to promote healthier lifestyles and reduce the burden of morbidity in the population.

Reviewer #2: Interesting topic. Well presented. Its surprising to see the average BMI below 30 over time. I suspect as people have decresead the use of tobacco, they have caught up on alcohol consumption. The advances in medicine likely explain the overall decreasing Mortality risk of the various rsik factors.

Reviewer #3: Firstly, it is commendable that the paper tackles pertinent public health matters and offers solutions to significant questions. Such an approach implies that the paper has practical implications and can contribute to the ongoing efforts to enhance public health.

Secondly, the paper's exploration of how risk factors, previously believed to impact health, have evolved over time is also noteworthy. It underscores the need for continually updating our comprehension of these risk factors that can affect morbidity and mortality, and adjusting our methodologies accordingly.

Lastly, the paper's identification of emerging risk factors such as mental health, social factors like insurance status, income level, literacy level, and substance use, which significantly impact overall morbidity and mortality at both the individual and population level, is valuable as it enables us to be proactive in addressing emerging public health concerns.

Overall, the paper seems to be informative and relevant to the public health field, providing insights that can help improve health outcomes.

6. PLOS authors have the option to publish the peer review history of their article (what does this mean?). If published, this will include your full peer review and any attached files.

Reviewer #1: No

Reviewer #2: No

Reviewer #3: **Yes: **Sakteesh V. Gurunathan

---

## [Author Response · Author response to Decision Letter 0]

4 May 2023

Dear Editor Venkata,

We thank you and the reviewers for the opportunity to revise and improve our manuscript. Please see our point by point responses to your and the reviewer comments below.

Thank you for your consideration,

Jen Kuk

Editor Comments

Comment - a) In methods section of abstract: Can you elaborate further on what specific methods were used?

Response – We have elaborated the methods to as follows:

“Data from the National Health and Nutrition Examination Surveys (NHANES III – 1988-1994 and NHANES 1999-2014; age ≥20 years) was used to examine differences in the odds ratio (OR) of 5-year mortality risk associated with various common health risk factors over the two survey periods using weighted logistic regression analysis adjusting for age, sex, obesity category and white ethnicity (n=28,279). “

Comment b) In methods section of the main article: Can you specify what kind of study this, currently you just mentioned that SPSS software was used, that we used ODDS ratio. May be a line indication what kind of observational study this is? 

Response – At the start of the methods we added this sentence: 

“The current study is a cross-sectional comparison of two nationally representative samples from 1988-1994 versus 1999-2014 with 5-year mortality follow-up.”

Comment c) At the end of the article, can we have separate brief conclusion section: include a take away point in the end

Response – We have added a title heading for our “Conclusions” section. This was our original conclusion paragraph. 

“This study demonstrates that developments in health care, social programs and the built environment may have altered how traditional risk factors relate to health and mortality over time. Thus, attention to not only the changing prevalence of traditional risk factors, but also examination of how these risk factors change in their association with morbidity and mortality is needed. “

- Apologies, but I am unclear if the second point is not sufficient for a “take away point”? I re-read the submission guidelines and a few recent publications and could not find reference to a formal ‘take away point’?

Comment d) Also may be a separate brief strength and limitation section?

Response – We have added a title heading for the “Strengths and Limitations” section

Journal Requirements

Comment 1. Please ensure that your manuscript meets PLOS ONE's style requirements, including those for file naming. The PLOS ONE style templates can be found at 

Response – We have revised the formatting of the main manuscript and author affiliations as stating in the documents.

Comment 2. We note that you have included the phrase “data not shown” in your manuscript. Unfortunately, this does not meet our data sharing requirements. PLOS does not permit references to inaccessible data. We require that authors provide all relevant data within the paper, Supporting Information files, or in an acceptable, public repository. Please add a citation to support this phrase or upload the data that corresponds with these findings to a stable repository (such as Figshare or Dryad) and provide and URLs, DOIs, or accession numbers that may be used to access these data. Or, if the data are not a core part of the research being presented in your study, we ask that you remove the phrase that refers to these data.

Response – We have removed our reference to unreported results and have cited published literature stating the same point instead (page 16-7).

Comment 3. Please include your full ethics statement in the ‘Methods’ section of your manuscript file. In your statement, please include the full name of the IRB or ethics committee who approved or waived your study, as well as whether or not you obtained informed written or verbal consent. If consent was waived for your study, please include this information in your statement as well. 

Response – Thank you. This is now included at the end of paragraph 1 on page 4.

Comment 4. Please review your reference list to ensure that it is complete and correct. If you have cited papers that have been retracted, please include the rationale for doing so in the manuscript text, or remove these references and replace them with relevant current references. Any changes to the reference list should be mentioned in the rebuttal letter that accompanies your revised manuscript. If you need to cite a retracted article, indicate the article’s retracted status in the References list and also include a citation and full reference for the retraction notice.

Response – I have reviewed the reference list and did not find any retractions for our citations.

Additional Editor Comments:

Good study, recommend edits as noted below

Comment - In abstract: In methods section, please describe more about what methods were used. Its too vague

Response – We have elaborated the methods to as follows:

“Data from the National Health and Nutrition Examination Surveys (NHANES III – 1988-1994 and NHANES 1999-2014; age ≥20 years) was used to examine differences in the odds ratio (OR) of 5-year mortality risk associated with various common health risk factors over the two survey periods using weighted logistic regression analysis adjusting for age, sex, obesity category and white ethnicity (n=28,279). “

Comment - In main article: In methods section, would it be possible to describe clearly what type of study this is. What kind of observational study this is? Cohort study?

Response – At the start of the methods we added this sentence: 

“The current study is a cross-sectional comparison of two nationally representative samples from 1988-1994 versus 1999-2014 with 5-year mortality follow-up.”

Comment - End of discussion, can we have a separate conclusion section and a separate strengths/limitation section?

Response – we have added heading to clearly note these sections.

Comment - After above revision. Manuscript will be ready for publication

Response – Thank you. We hope that our revisions have adequate addressed your concerns and we thank you for your contribution in this process.

5. Review Comments to the Author

Reviewer #1: The authors have provided an analysis and reviewed data on the prevalence of various health risk factors over a period of nearly three decades in order to identify patterns in population health. By analyzing trends in the prevalence of these risk factors, they likely sought to determine how the health of the population has changed over time.

The examples of health risk factors that the authors have analyzed include smoking, alcohol consumption, physical inactivity, poor nutrition, and obesity amongst other social, physiological risk factors.

By examining changes in the prevalence of these risk factors over time, the authors have been able to identify trends and patterns that could inform public health policy and interventions aimed at improving population health.

Overall, it appears that the authors have conducted a comprehensive analysis of health risk factors over a long period of time in order to gain insights into population health patterns. The findings of this analysis could have important implications for efforts to promote healthier lifestyles and reduce the burden of morbidity in the population.

Response – We thank the reviewer for their comments and for taking the time to review our manuscript.

Reviewer #2: Interesting topic. Well presented. Its surprising to see the average BMI below 30 over time. I suspect as people have decresead the use of tobacco, they have caught up on alcohol consumption. The advances in medicine likely explain the overall decreasing Mortality risk of the various rsik factors.

Response – Thank you. We completely agree with you.

Reviewer #3: Firstly, it is commendable that the paper tackles pertinent public health matters and offers solutions to significant questions. Such an approach implies that the paper has practical implications and can contribute to the ongoing efforts to enhance public health.

Secondly, the paper's exploration of how risk factors, previously believed to impact health, have evolved over time is also noteworthy. It underscores the need for continually updating our comprehension of these risk factors that can affect morbidity and mortality, and adjusting our methodologies accordingly.

Lastly, the paper's identification of emerging risk factors such as mental health, social factors like insurance status, income level, literacy level, and substance use, which significantly impact overall morbidity and mortality at both the individual and population level, is valuable as it enables us to be proactive in addressing emerging public health concerns.

Overall, the paper seems to be informative and relevant to the public health field, providing insights that can help improve health outcomes.

Response – Thank you. We appreciate the reviewer’s kind words and for reviewing our manuscript.

---

## [Editor Report · Decision Letter 1]

22 May 2023

Is anyone truly healthy? Trends in health risk factors prevalence and changes in their associations with all-cause mortality

PONE-D-23-10979R1

Dear Dr. Kuk,

We’re pleased to inform you that your manuscript has been judged scientifically suitable for publication and will be formally accepted for publication once it meets all outstanding technical requirements.

Kind regards,

Vikramaditya Samala Venkata

Academic Editor

PLOS ONE

Additional Editor Comments (optional):

Thank you for making all the required changes. Including explaining the methods, type of study and adding conclusions and limitations. Your article will surely help the medical community.
---

## [Editor Report · Acceptance letter]

24 May 2023

PONE-D-23-10979R1 

Is anyone truly healthy? Trends in health risk factors prevalence and changes in their associations with all-cause mortality. 

Dear Dr. Kuk:

I'm pleased to inform you that your manuscript has been deemed suitable for publication in PLOS ONE. Congratulations! Your manuscript is now with our production department. 

Kind regards, 

on behalf of

Dr. Vikramaditya Samala Venkata 

Academic Editor

PLOS ONE